# Endophytic Fungal and Bacterial Microbiota Shift in Rice and Barnyardgrass Grown under Co-Culture Condition

**DOI:** 10.3390/plants11121592

**Published:** 2022-06-16

**Authors:** Shuyan Li, Qiling Yan, Jieyu Wang, Qiong Peng

**Affiliations:** 1Longping Branch, Graduate School of Hunan University, Changsha 410125, China; lishuyan202205@163.com (S.L.); yanqiling17017@163.com (Q.Y.); wangjieyu202205@163.com (J.W.); 2Hunan Agricultural Biotechnology Research Institute, Hunan Academy of Agricultural Sciences, Changsha 410125, China

**Keywords:** rice, barnyardgrass, biotic stress, endophytic bacteria, endophytic fungi, PICRUSt, FUNGuild, network

## Abstract

Although barnyardgrass (*Echinochloa crus-galli* L.) is more competitive than rice (*Oryza sativa* L.) in the aboveground part, little is known about whether barnyardgrass is still competitive in recruiting endophytes and the root microbiota composition variation of rice under the barnyardgrass stress. Here, by detailed temporal characterization of root-associated microbiomes of rice plants during co-planted barnyardgrass stress and a comparison with the microbiomes of unplanted soil, we found that the bacterial community diversity of rice was dramatically higher while the fungal community richness was significantly lower than that of barnyardgrass at BBCH 45 and 57. More importantly, rice recruited more endophytic bacteria at BBCH 45 and 57, and more endophytic fungi at BBCH 17, 24, 37 to aginst the biotic stress from barnyardgrass. Principal coordinates analysis (PCoA) showed that rice and barnyardgrass had different community compositions of endophytic bacteria and fungi in roots. The PICRUSt predictive analysis indicated that majority of metabolic pathways of bacteria were overrepresented in barnyardgrass. However, eleven pathways were significantly presented in rice. In addition, rice and barnyardgrass harbored different fungal trophic modes using FUNGuild analysis. A negative correlation between bacteria and fungi in rice and barnyardgrass roots was found via network analysis. Actinobacteria was the vital bacteria in rice, while Proteobacteria dominated in barnyardgrass, and Ascomycota was the vital fungi in each species. These findings provided data and a theoretical basis for the in-depth understanding of the competition of barnyardgrass and endophytes and have implications relevant to weed prevention and control strategies using root microbiota.

## 1. Introduction

Barnyardgrass (*Echinochloa crus-galli* L.) is one of the most troublesome weeds in transplanted and direct-seeded rice worldwide [1], which represents an important biological constraint on rice productivity [2]. Barnyardgrass is a C4 plant, which has the characteristics of high photosynthetic rate and high utilization efficiency of water and nitrogen resources [3]. It is known that the leaf photosynthetic rate, root biomass, root oxidation activity, and dry matter accumulation of rice are significantly decreased in the presence of barnyardgrass co-culture [4].

Plant roots are colonized by complex microbiota. Plants can secrete signal chemicals from roots to induce defense metabolites against competitors [5], and these chemicals can change the soil microbiota, especially endophytes [6]. Compared with rhizosphere microorganisms, endophytic microorganisms may show greater potential in phytoremediation [7]. Endophytes include both bacteria and fungi that colonize the plant tissues either intracellularly or intercellularly without causing any harm to the plant [8]. Endophytes can establish mutually beneficial relationships with plants, enabling them to better resist biotic and abiotic challenges [9,10].

The beneficial effects of endophytes on rice growth are mediated by several functional traits, including phytohormones and siderophores’ production, nitrogen fixation, and metal detoxification [10]. In rice, various types of endophytes have been isolated and characterized for their role in promoting rice health and vigor under biotic and abiotic stress conditions [11]. *Paecilomyces formosus LWL1* and *Bacillus pumilus* can enhance rice growth under heat and high salt stresses [12,13]. Rice blast disease was reduced by *Streptomyces hygroscopicus* OsiSh-2 in [14]. Researchers found that barnyardgrass infestation and mixed culture could alter the microbiological properties of the soil, especially microbial biomass and community structure [15]. Therefore, changes in the microbial community composition induced by barnyardgrass infestation may generate the feedback on rice growth and reproduction in the paddy system.

As barnyardgrass infestation problems become more frequent, it will be important to gain detailed knowledge of the consequences of invasive barnyardgrass on rice-associated microbiomes. Here, we conducted a detailed temporal profiling of the endosphere communities of rice plants grown with barnyardgrass stress to investigate the following questions: (1) whether barnyardgrass is more competitive than rice in the root microbiome in controlled microbial sources, and (2) how endophytes shift in two species. The results provide insights into the impacts of barnyardgrass on rice root microbiota establishment and the control of barnyardgrass from the perspective of the interaction of microorganisms.

## 2. Materials and Methods

### 2.1. Plant Growth 

The experiments and analysis were conducted at the experimental farm of Hunan Academy of Agricultural Sciences. The experiments were conducted during the summer of 2020 to study the differences of endophytes in rice and barnyardgrass, and how endophytes improved the ability of rice to compete with the co-planted barnyardgrass at different growth stages of rice. For this, rice (“Tianyou Huazhan”) and barnyardgrass were cultivated together at a 1:1 co-planted density (two rice plants and two barnyardgrass plants were planted together at each pot). At each stage, each pot was repeated six times. Rice “Tianyou Huazhan” was provided by Hunan Rice Research Institute, and barnyardgrass was collected from paddy fields in Hunan province.

To avoid seed endophytes and surface-related microorganisms, rice and barnyardgrass seeds were dehulled, and surface-sterilized in 75% ethanol for 30 s and 2.5% sodium hypochlorite 3 times for 15 min, then seed germinated in MS agar media. After germination, the rice and barnyardgrass (at BBCH 07) were transplanted to the plastic pots. The pots had dimensions of 46 cm (length) × 35 cm (width) × 17 cm (height), filled with 15.0 kg of sieved (2 mm) dry paddy soil. The soil was derived from the same field block that was only used for rice cultivation. Seedlings were watered and fertilized as required according to conventional field management practices. Other weeds were manually pulled out to prevent additional competition. 

### 2.2. Sample Collection

The root samples were collected at BBCH 17, 24, 37, 45, and 57 [16]. Ten rice plants selected from each growth stage were sampled for root endophytes’ analysis.

Excess soil was gently shaken from the roots and the roots were washed with flowing water. The roots were cleaned up with sterilized scissors and transferred to a sterile tube with 50 mL of chilled (4 °C) sterile phosphate buffered saline (PBS) solution and shaken vigorously until no soil remained on the root surface. Then, the samples were surface-sterilized in 75% ethanol for 90 s and washed in sterile water 3 times. After washing, roots were dried on sterile filter paper, wrapped in sterile aluminum foil, flash-frozen in liquid N_2_, and stored at −80 °C. 

### 2.3. DNA Extraction and PCR Amplification 

Microbial genomic DNA was extracted from rice root samples using the E. Z. N. A.^®^ Soil DNA Kit (Omega Bio-tek, Norcross, GA, USA) according to the manufacturer’s protocols. The DNA extraction quality was analyzed using 1% agarose gel, and DNA concentration and purity were determined using the NanoDrop 2000 UV-vis spectrophotometer (Thermo Scientific, Wilmington, NC, USA).

The V3–V4 regions of the bacteria 16S ribosomal RNA gene from rice root samples were amplified with the primer pair 779F(5′-AACMGGATTAGATACCCKG-3′) 1193R(5′-ACGTCATCCCCACCTTCC-3′), and the hypervariable region of ITS was amplified with the primer pair ITS1F(5′-CTTGGTCATTTAGAGGAAGTAA-3′) ITS2R (5′-GCTGCGTTCTTCATCGATGC-3′) using an ABI GeneAmp^®^ 9700 PCR thermocycler (ABI, CA, USA).

The PCR thermal cycle was as follows: initial denaturation at 95 °C for 3 min, followed by 27 cycles of denaturing at 95 °C for 30 s, annealing at 55 °C for 30 s, extension at 72 °C for 45 s, single extension at 72 °C for 10 min, and end at 4 °C. 

For bacteria, the formal PCR test was performed using the TransGen AP221-02: TransStart Fastpfu DNA Polymerase. PCR reactions were performed in triplicate as 20 μL mixtures containing 4 μL of 5× FastPfu Buffer, 2 μL of 2.5 mM dNTPs, 0.8 μL of each primer (5 μM), 0.4 μL of FastPfu Polymerase, 0.2 μL of BSA, and 10 ng of template DNA. PCR was performed using the ABI GeneAmp^®^ 9700 (Bio-Rad, USA) with the following procedure: 95 °C for 3 min, followed by 27 cycles at 95 °C for 30 s, 55 °C for 30 s, 72 °C for 45 s, a final extension at 72 °C for 10 min, and end at 10 °C.

For fungi, using TaKaRa rTaq DNA Polymerase, PCR reactions were performed in triplicate in 20 μL mixtures containing 2 µL of 10× Buffer, 2 µL of 2.5 mM dNTPs, 0.8 μL of each primer (5 μM), 0.2 µL of rTaq Polymerase, 0.2 μL of BSA, and 10 ng of template DNA. PCR was performed using the ABI GeneAmp^®^ 9700 (Bio-Rad, Waltham, Massachusetts, USA) with the following procedure: 95 °C for 3 min, followed by 27 cycles at 95 °C for 30 s, 55 °C for 30 s, 72 °C for 45 s, a final extension at 72 °C for 10 min, and end at 10 °C. The PCR products were evaluated using 2% (wt./vol) agarose gel electrophoresis.

### 2.4. Sequence Analysis

Sequencing and library construction were performed by Beijing Biomarker Technologies Co. The raw 16S rRNA gene sequencing reads were demultiplexed, quality-filtered using the fastp version 0.20.0 [17], and merged using FLASH version 1.2.7 [18] with the following criteria: (i) The 300 bp reads were truncated at any site receiving an average quality score of <20 over a 50 bp sliding window, and the truncated reads shorter than 50 bp were discarded. Reads containing ambiguous characters were also discarded. (ii) Only overlapping sequences longer than 10 bp were assembled according to their overlapped sequence. The maximum mismatch ratio of the overlap region was 0.2. Reads that could not be assembled were discarded. (iii) Samples were distinguished according to the barcode and primers, and the sequence direction was adjusted, exact barcode matching, 2 nucleotide mismatch in primer matching.

Operational taxonomic units (OTUs) with a 97% similarity cutoff [19,20] were clustered using UPARSE version 7.1 [20], and chimeric sequences were identified and removed. The taxonomy of each OTU representative sequence was analyzed using the RDP Classifier version 2.2 [21] against the 16S rRNA database (Silva v138) and the ITS database (unite 8.0) using a confidence threshold of 0.7.

### 2.5. Statistical Analysis

The α-diversity consisted of the ACE estimator and Shannon and Coverage species diversity indices. These were carried out using Mothur version 1.30 (http://www.mothur.org/ (accessed on 6 April 2022). Additionally, β-diversity estimators were used to analyze the differences among all samples. Principal coordinate analysis (PCoA) was performed based on the Bray–Curtis distances among the bacterial and fungal communities. Bacterial and fungal taxonomic distributions of sample communities were visualized using the R package software. The Mann–Whitney U test was used to examine differences in bacterial and fungal composition between the two groups. The differences in diversity and richness among treatments were determined using one-way ANOVA (SPSS 22.0). The bacterial functions were predicted by PICRUSt version 1.1.0 (http://picrust.github.io/picrust/ (accessed on 8 April 2022). The fungal functions were predicted by FUNGuild version 1.0 (http://www.funguild.org/ (accessed on 9 April 2022). The influence of communities on one another was assessed using network analysis (Python2.7, Networkx, and Gephi).

## 3. Results

### 3.1. Soil Physicochemical Analysis

The contents of soil organic matter, total nitrogen, total phosphorus, total potassium, effective nitrogen, effective phosphorus, and effective potassium were 28.37 g/kg, 1.67 g/kg, 0.85 g/kg, 17.40 g/kg, 131.00 mg/kg, 2.30 mg/kg, and 150.00 mg/kg, respectively. The soil pH was 6.2. The cation exchange capacity was 98.80 us/cm. The electroconductivity was 14.40 cmol/kg. In addition, the soil texture was sandy loam.

### 3.2. Sequencing Results

A total of 7,394,465 high-quality sequences were retrieved after denoising, merging, and chimera checking. After separating bacterial and fungal reads, 3,494,443 bacterial sequences remained, and a total of 3759 OTUs were obtained, belonging to 46 phyla, 137 classes, 294 orders, 507 families, 983 genera, and 1801 species. For fungal sequences (3,900,022), a total of 1796 OTUs were obtained, belonging to 11 phyla, 41 classes, 108 orders, 241 families, 471 genera, and 750 species. 

### 3.3. Alpha Diversity and Community Composition in Soil Microbiota

Table 1 shows that the bacterial community diversity (Shannon index), richness (Chao index), and coverage (Coverage index) had no difference in soil samples, and similar trends for fungi. According to the community composition analysis, for bacteria, there were 8 main classes, including Acidobacteriae, Actinobacteria, Alphaproteobacteria, Bacilli, Gammaproteobacteria, Thermoleophilia, TK10, and MB-A2-108. and there were 5 main fungal classes, namely, Agaricomycetes, Sordariomycetes, Dothideomycetes, Eurotiomycetes, and Rozellomycotina_cls_Incertae_sedis. Coupled with the bacterial (Appendix A) and fungal community composition (Appendix A), we concluded that the composition of microorganisms in the soil was roughly the same.

### 3.4. Alpha- and Beta-Diversity in Rice Roots Endophytic Communities

Due to the competition between rice and barnyardgrass, differences were observed in alpha-diversity indices for bacterial and fungal communities among the two species (Figure 1). More specifically, the bacterial community richness (Sobs index) of rice was significantly lower than that of barnyardgrass at BBCH 37 and 45. The bacterial community diversity (Shannon index) of rice was significantly lower than that of barnyardgrass at BBCH 17, while it was dramatically higher than barnyardgrass at BBCH 24 and 45. When it comes to fungi, those parameters showed a different trend. No differences could be seen for the fungal community diversity. However, the fungal community richness of rice was significantly lower than that of barnyardgrass at BBCH 45 and 57.

Beta-diversity analysis (Figure 2) revealed the similarities or differences in the composition of the bacterial and fungal communities of the two species. PCoA based on Bray–Curtis distance was used to study the difference in community composition of endophytes in rice and barnyardgrass roots. 

In PCoA of Bray–Curtis distance from all samples, the distance in the ordination plot for the rice and barnyardgrass samples showed differences in bacterial communities, and the largest distance could be seen at the reproductive stage (BBCH 45, 57). For fungi, a significant difference occurred at each period. The results indicated that the community composition of endophytes between rice and barnyardgrass roots showed dramatic differences.

### 3.5. Endophytic Bacterial and Fungal Species Composition

#### 3.5.1. Bacterial Community Composition

At the class level (Figure 3a), all the libraries, regardless of rice and barnyardgrass, were composed mainly by Gammaproteobacteria and Alphaproteobacteria, with minor amounts of Actinobacteria and Myxococcia. Alphaproteobacteria and Actinobacteria were more frequent in rice, whereas Gammaproteobacteria and Myxococcia were much more abundant in barnyardgrass. 

At the genus level, at BBCH 17 (Figure 4a), apart from Proteobacteria (p), the majority of bacteria were significantly enriched in barnyardgrass, including *Stenotrophomonas*, Rhodocyclaceae(f), *Sideroxydans*, *Acidovorax*, *Azospira*, *Bradyrhizobium*, *Ideonella*, *Methylocystis*, Gallionellaceae(f), Rhodocyclaceae(f), *Pleomorphomonas*, Burkholderiales(o), *Haliangium*, and *BBMC-4*.

At BBCH 24 (Figure 4b), 8 genera were enriched in rice, namely, *Acidovorax*, *Microvirga*, *Bradyrhizobium*, *Streptomyces*, Rhizobiaceae(f), *Delftia*, *Burkholderia-Caballeronia-Paraburkholderia*, and *Paenibacillus*. Whereas 7 genera were much more abundant in barnyardgrass, including Rhodocyclaceae(f), *Sideroxydans*, Gallionellaceae(f), Bacteroidales(o), Hungateiclostridiaceae(f), *Massilia*, and Burkholderiales(o).

At BBCH 37 (Figure 4c), apart from *Corynebacterium*, more genera were enriched in barnyardgrass, namely *Sideroxydans*, Gallionellaceae(f), *Massilia*, *Phyllobacterium*, BIrii41(f), *Paludibacter*, Bacteroidetes_vadinHA17(f), *Clostridium_sensu_stricto_10*, *Barnesiellaceae*, Deinococcus(f), *Turneriella*, Dysgonomonadaceae(f), *Micrococcus*, and Hungateiclostridiaceae(f).

At BBCH 45 and 57, more genera were enriched in rice. To be specific, at BBCH 45 (Figure 4d), *Bradyrhizobium*, *Ralstonia*, *Clostridium_sensu_stricto_1*, *Rhodococcus*, Streptomyces, Delftia, Paenibacillus, Microvirga, Methylobacter, Pseudomonas, Methylocystis, and 67-14(f) were more frequent in rice, and only *Anaeromyxobacter*, SB-5(f), and *Derxia* were enriched in barnyardgrass. At BBCH 57 (Figure 4e), *Ciceribacter*, *Ralstonia*, *Bradyrhizobium*, *Rhodococcus*, Rhizobiales_Incertae_Sedis(f), *Allorhizobium-Neorhizobium-Pararhizobium-Rhizobium*, Rhizobiaceae(f), *Schlegelella*, OPB41(o), and *Phenylobacterium* were much more abundant in rice, while Rhodocyclaceae(f), *Anaeromyxobacter*, Burkholderiales(o), Bacteroidales(o), and *Azospira* were enriched in barnyardgrass.

Overall, barnyardgrass was more competitive than rice in recruiting endophytic bacteria in the vegetative growth stage (BBCH 17, 24, 37), while rice was able to recruit more bacteria at the reproductive growth stage (BBCH 45, 57).

Data are represented as relative abundance (%) of genus in each group. Statistical analyses were performed with the Mann–Whitney U test between the rice and barnyardgrass groups. * *p* < 0.05, ** *p* < 0.01, the rice group vs. the barnyardgrass group. 

#### 3.5.2. Fungal Community Composition

At the class level (Figure 3b), all the libraries, regardless of rice and barnyardgrass, were composed by Sordariomycetes, Eurotiomycetes, Tremellomycetes, and Saccharomycetes, whereby the former two classes were frequent in rice, whereas the latter two were much more frequent in barnyardgrass.

At BBCH 17 (Figure 5a), apart from *Acrophialophora*, the majority of genera were much more abundant in rice, including *Thermomyces*, *Alternaria*, *Cladosporium*, *Solicoccozyma*, *Fusarium*, *Tausonia*, Glomerellales(o), Pleosporaceae(f), *Filobasidium*, *Vishniacozyma*, and *Tetracladium.*

At BBCH 24 (Figure 5b), *Acrophialophora* was significantly enriched in rice roots. At BBCH 37 (Figure 5c), Fungi(k), Ascomycota(p), *Aspergillus*, *Penicillium*, and *Apodus* were much more frequent in barnyardgrass. At BBCH 45 (Figure 5d), Sordariales(o) and *Cladosporium* were frequent in rice, while Glomeraceae(f), *Exserohilum*, and *Paraphoma* were more abundant in barnyardgrass. At BBCH 57 (Figure 5e), *Phaeosphaeriopsis*, Chytridiomycota(p), Didymellaceae(f), *Epicoccum*, Helotiales(o), *Phoma*, *Albifimbria*, *Thielavia*, Tremellaceae(f), *Ascobolus*, *Acremonium*, Glomerellales(o), *Pseudopithomyces*, and *Scolecobasidium* were significantly enriched in barnyardgrass.

Overall, at BBCH 17, 24, and 37, rice was more competitive than barnyardgrass in recruiting fungi, while the competition of barnyardgrass gradually increased at BBCH 45 and 57.

In conclusion, the competition between the two species for bacteria and fungi showed totally opposite trends. To be more specific, at the vegetative growth stage, barnyardgrass had a strong ability to recruit bacteria, while its ability to recruit fungi was lower than that of rice. At the reproductive growth stage, rice was more competitive in recruiting bacteria, but less so for fungi.

### 3.6. Bacterial and Fungal Metabolic Pathways differentially Represented in Rice and Barnyardgrass

Possible bacterial metagenomes were estimated using PICRUSt, with GreenGenes as the underlying database. The differences were significant in two species at different growth stages. There were 11 level 3 pathways overrepresented in rice (Figure 6), while the most metabolic pathways were remarkably presented in barnyardgrass. 

To be more specific, 11 level 3 pathways were overrepresented in rice in each growth period. Two of them belonged to energy metabolism (photosynthesis, oxidative phosphorylation), and the remaining ones came from carbohydrate metabolism (citrate cycle (TCA cycle)), amino acid metabolism (lysine biosynthesis), metabolism of other amino acids (D-glutamine and D-glutamate metabolism), biosynthesis of other secondary metabolites (isoquinoline alkaloid biosynthesis), glycan biosynthesis and metabolism (peptidoglycan biosynthesis), membrane transport (bacterial secretion system), metabolism of terpenoids and polyketides (zeatin biosynthesis), nucleotide metabolism (pyrimidine metabolism), as well as xenobiotics biodegradation and metabolism (toluene degradation).

Majority of the pathways were overrepresented in barnyardgrass (Appendix A), including amino acid metabolism, biosynthesis of other secondary metabolites, carbohydrate metabolism, energy metabolism, glycan biosynthesis and metabolism, membrane transport, metabolism of other amino acids, metabolism of terpenoids and polyketides, and xenobiotics biodegradation and metabolism.

In the present study, PICRUSt predicted that rice resisted barnyardgrass stress by lifting the relative abundance of 11 metabolic pathways. Although, a myriad of metabolic pathways were significantly represented in barnyardgrass.

FUNGuild was used to predict the nutritional and functional groups of the fungal communities under different treatments. There were eight trophic mode groups (Figure 7a), among which Pathogen-Saprotroph-Symbiotrophthe and Pathotroph were the primary trophic modes in barnyardgrass, while the rice samples primarily harbored Pathotroph-Saprotroph-Symbiotroph and Saprotroph. 

Eight trophic mode groups could be classified, with Pathotrophs, Saprotrophs, and Symbiotrophs being the major components. The saprotroph (Figure 7c) composition analysis showed that the relative abundance of dung saprotrophs was significantly higher in rice, which also harbored more soil saprotroph, undefined saprotroph, and wood saprotroph, but no dramatic difference. For barnyardgrass, the main trophic mode group was Pathotroph, and the pathotroph composition (Figure 7b) showed that the relative abundance of plant and animal pathogens was higher in barnyardgrass. Appendix A shows that the fungi showed pathogenicity in barnyardgrass. 

In addition, the symbiotrophic (Figure 7d) mode groups primarily consisted of arbuscular mycorrhizal, endophyte, and ectomycorrhizal. The relative abundance of arbuscular mycorrhizal and ectomycorrhizal in rice was significantly larger than that observed in the barnyardgrass groups.

### 3.7. Interactions between Endophytic Bacterial and Fungal Communities

To unravel the relationship between endophytic fungi and bacteria, we constructed a co-occurrence network for each species. Network analysis (Figure 8 and Figure 9) indicated that there was a significant influence of the bacterial community on the fungal one, and vice versa. In other words, the fungal community might be predicted based on the bacterial one. The relationship between most bacterial and fungal communities was negative, regardless of species. 

The species with significant differences (*p* < 0.05) were selected, and correlation coefficients, such as Spearman ranks, were calculated between species to reflect the correlation between species. The size of the node in Figure 8 and Figure 9 represents the species abundance, the yellow node represents bacteria, and the green node indicates fungi. The color of the line indicates positive or negative correlation, where red indicates a positive correlation and green indicates a negative correlation. The thickness of the line indicates the size of the correlation coefficient, and the thicker the line, the higher the correlation between species. The more lines, the closer the connection between the nodes.

To reveal the key microbes in the networks, we examined the nodes with maximum (max) degree, max closeness, and max betweenness centrality. The larger the three measures are, the more important the node is in the network. Nodes with max degree and betweenness were the nodes that had the most interactions with other nodes [22,23]. Moreover, nodes with the highest betweenness could serve as brokers [23,24]. The largest closeness means that the nodes were closer to the center of the network. These nodes were important in maintaining the bacteria–fungi interactions in the networks. Appendix A show the key bacteria and fungi in each species and their taxa in networks. 

Members in phylum Actinobacteria (e.g., genus Delftia, Pseudomonas) and Ascomycota (e.g., Glomerellales (o), Thermomyces) were important taxa in networks from rice. While for barnyardgrass, members in Proteobacteria (Methylocystis, Pseudomonas) and Ascomycota (Pseudopithomyces, Apodus, Acrophialophora) played a vital role in the network.

The results indicated that the disturbance of barnyardgrass reduced the synergistic effect between fungi and bacteria, thus interfering with the growth and development of rice.

## 4. Discussion

### 4.1. Barnyardgrass mainly Affected the Endophytic Bacteria in Rice 

Endophytes may contribute to the interactions between plant species. In our work, we have observed significant differences about the diversity and richness of endophytes in rice and barnyardgrass roots. The richness of endophytic fungi classified to the genus level in our study was lower than that of bacteria in both species. This result was consistent with the previous studies, which revealed that the ratio of bacteria to fungi was significantly higher in the rhizosphere [15], and fungi were less sensitive than bacteria when co-cultured with barnyardgrass [22]. We also found that the fungal community richness of rice was significantly lower than that of barnyardgrass at BBCH 45 and 57. The reason may be that the fungal community was related to the change of soil nutrient status [23], and barnyardgrass secreted a series of root exudates, so the nutrient composition near the barnyardgrass root system changed according to its metabolites [15]. Coupled with the Beta-diversity analysis, we deduced that the competition between barnyardgrass and rice was mainly reflected in endophytic bacteria, rather than fungi. 

### 4.2. Rice Relied on Fungi at Vegetative Stage and Bacteria at Reproductive Stage against Barnyardgrass Stress 

Betaproteobacteria, Firmicutes, and Gammaproteobacteria were the main classes in rice roots [24]. Gammaproteobacteria could be related to carbon fixation [25]. However, in our study, Gammaproteobacteria was much more abundant in barnyardgrass. The reason might be that barnyardgrass was more competitive than rice in recruiting carbon-fixing bacteria to complete growth and development as C4 plants.

Vegetative growth is an important growth phase in many crops as it determines the amount of biomass production. A strong vegetative growth of rice plants reflected a higher plant height and greater plant biomass, and a larger number of tillers and panicles [26,27]. In our study, at the vegetative stage, we found that barnyardgrass was more competitive in recruiting bacteria, while rice recruited more fungi. We deduced that barnyardgrass exuded a wide variety of metabolites to inhibit the growth of rice, which would need various enzymes to be utilized by bacteria [28]. This trend was totally opposite at the reproductive stage. The reason may be that species modified the rhizospheric microbial community structure to generate positive feedback [22,29]. 

Biological nitrogen fixation is carried out by microorganisms utilizing complex procedures with the help of enzymes, such as nitrogenase, that effectively converted atmospheric nitrogen into ammonia. Nitrogenase is a highly conserved protein, and this enzyme was commonly found in all nitrogen-fixing bacteria, and a series of diazotrophic bacteria was detected in rice roots [30]. At the vegetative growth stage, we found that the relative abundance of bacteria with nitrogen-fixing and nitrogen-cycling effects was higher in barnyardgrass, such as *Azospira*, *Bradyrhizobium*, *Stenotrophomonas*, and *Allorhizobium-Neorhizobium-Pararhizobium-Rhizobium*. The previous studies also showed that a strong shift appeared in the rice diazotrophic under abiotic stress [27]. Under the strong pressure of barnyardgrass, rice also recruited more fungi to complete growth, including *Cladosporium*, *Solicoccozyma*, *Fusarium*, and *Tausonia*. These fungi may play a vital role in resisting biotic stress from barnyardgrass. However, further study is needed to validate the colonization of rice by these fungi. 

While, at the reproductive stage (BBCH 45, 57), rice was more competitive to endophytic bacteria than barnyardgrass, especially for nitrogen-fixing bacteria (*Bradyrhizobium*, *Microvirga*, *Methylobacter*, *Ciceribacter*), under the barnyardgrass stress, rice needs to recruit more nitrogen-fixing bacteria. Another reason may be that rice secreted a series of root exudates which provided nutrition for bacterial colonization. At the same time, barnyardgrass also recruited some fungi to resist the competition of rice, such as *Epicoccum spp.,* which can produce many secondary metabolites such as polyketides, polyketide hybrids, and diketopiperazines, several of which present biological activities, such as antimicrobial and antioxidant activity [31]. 

### 4.3. Rice Utilized Metabolism Pathways and Different Trophic Modes to Improve Its Defensive System

In our study, the majority of metabolic pathways were upregulated in barnyardgrass, namely pathways related to carbohydrate metabolism, e.g., starch and sucrose metabolism, which could produce energy, a carbon source, and various compounds for bacterial metabolism. This may be caused by the greater diversity of complex carbohydrates in barnyardgrass roots that would need various enzymes to be utilized by bacteria [28]. Other researchers also found that activities of key enzymes involved in sucrose-to-starch conversion in rice root were dramatically reduced when grown with barnyardgrass [32]. Likewise, membrane transport (PTS) had a positive effect on the uptake and metabolism of carbohydrates, nitrogen and phosphorus utilization, and resistance to certain pathogens [33]. It was known that endophytes escaped plants’ defense mechanisms by activating genes involved in glutathione metabolism. Glutathione (or a functionally homologous thiol) is an essential metabolite with multiple functions in plants, especially for defense and detoxification [34]. Carotenoid pigments are essential for photosynthetic growth in higher plants and protection against photooxidation [35]. The endophytic bacteria may promote the plant growth by increasing the carotenoid content in the host [36]. 

However, 11 level 3 pathways were overrepresented in rice, namely, pathways related to energy metabolism, such as the oxidative phosphorylation (OXPHOS) system, which consists of the mitochondrial respiratory pathway and the ATP synthase, and is the central energy-producing pathway [37]. The citric acid cycle is the final common oxidative pathway for carbohydrates, fats, and amino acids, and it is the most important metabolic pathway for the energy supply to the organism [38]. However, why the genes involved in the rest of the pathways are apparently more common in rice roots remains to be elucidated. 

Additionally, trophic status was used to further explain how the fungal community helped rice to against barnyardgrass stress. Majority of the predicted fungi observed in rice belonged to Saprotroph and Pathotroph-Saprotroph-Symbiotroph, and the rice harbored more Pathotroph-Saprotroph-Symbiotroph compared with barnyardgrass. A previous study found that when infested by fusarium wilt disease, the relative abundance of Pathotroph-Saprotroph-Symbiotroph of cucumber was upregulated [39]. Saprotrophic fungi was a key microbial group mediating nutrient cycling in the terrestrial ecosystem by decomposing complex organic materials via the release of lignocellulolytic enzymes [40]. The higher fungi proportion of Saprotroph and Pathotroph-Saprotroph-Symbiotroph involved in the rice indicated that rice competed against infestation of barnyardgrass by enhancing saprotrophic growth. While, for barnyardgrass, majority of the predicted fungi belonged to Pathotroph and Pathogen-Saprotroph-Symbiotroph, which indicated that barnyardgrass competed against infestation of rice by enhancing Pathotrophic growth. The relative abundance of plant and animal pathogens was higher in barnyardgrass. Fungal plant pathogen species fall into two main categories: one involving biotrophic pathogens, which form close interactions with host plants and can persistently utilize living tissue (biotrophs), and the other including necrotrophic pathogens, which kill tissue to extract nutrients (necrotrophs) [41,42]. 

### 4.4. The Fungal Community May Be Predicted Based on the Bacterial One

Prior studies on the endophytic community in rice focused either on bacteria or fungi. However, as no creature thrives in isolation, the interaction of bacterial and fungal communities in different plant species can be of great importance [23]. Bacterial and fungal communities played different roles in regulating plant growth and health. The influence of the bacterial community on the fungal community in two species was found by network analysis. 

Bacteria may play a positive role in fungal activity through producing cellulases and pectinases, which increase available substrates for fungi. In addition, bacteria may also break down solutes that are toxic to certain fungi [43] and may increase the amount of nitrogen available to them [44]. However, bacteria also exert antifungal strategies by secreting a series of inhibitory factors such as HCN, antibiotics, lytic enzymes, and volatiles, as well as nutrient-sequestering factors such as iron-chelating siderophores [45,46,47]. When the density of saprophytic fungi in the rhizosphere increases, the enrichment of bacteria with antifungal properties also increases, such as those secreting siderophores, cyanide, and lyase [48]. Fungi also influence bacteria by releasing beneficial nutrients and using fungal hyphae in helping bacterial transport to sites that could not be reached by bacterial cells alone [49].

In our study, we found that *Pseudomonas* was vital in each species. *Pseudomonas* was important in rice, which can live in soil and colonize the rice root surfaces, and utilize soil or rhizosphere carbon sources for fixing nitrogen [50]. *Delftia* also played an important role in rice against biotic stress. The previous study found that *Delftia*, a betaproteobacterium, was characterized as a plant growth-promoting bacterium with a ‘helper’ function, enhancing the performance of rhizobial inoculant strains during the co-inoculation of plants [37].

We deduced that rice mainly competes with barnyardgrass for nitrogen-fixing microbes to accomplish its growth, and nitrogen-fixing microbes possibly formed the defense system of rice against biotic stress from barnyardgrass. However, the exact mechanism and direction of the two communities’ influence on one another remains to be elucidated and would require further studies, such as meta-transcriptomic ones. 

## 5. Conclusions

Rice and barnyardgrass showed totally opposite trends in recruiting endophytes. Specifically, at the vegetative growth stage, barnyardgrass had a strong ability in recruiting bacteria, while its ability to recruit fungi was lower than that of rice. At the reproductive growth stage, rice was more competitive in recruiting bacteria, but less so for fungi. In general, the strong competitiveness of barnyardgrass was significant, in terms of the relative abundance of endophytes, metabolic pathways, and the synergistic effect between fungi and bacteria. Moreover, the competition between them is mainly reflected in the nitrogen-fixing microbes. To illustrate the cascading effects of plant–soil–microbiome interactions in improving the ability of rice to resist barnyardgrass stress, further study is essential to isolate root microbiota and identify their function. More importantly, further study is required to illustrate how plants shape their microbiota, whether by root exudates, root structure, or another more complex mechanism.

## Figures and Tables

**Figure 1 plants-11-01592-f001:**
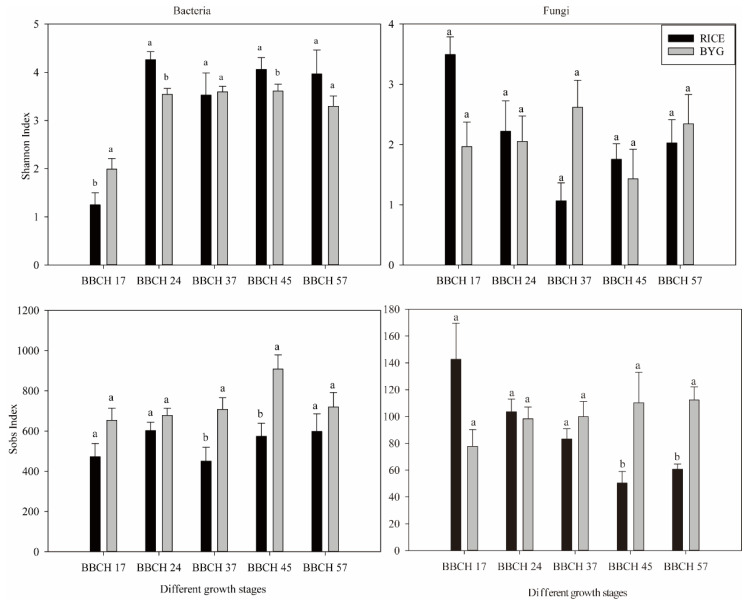
Species diversity (Shannon index) and richness (Sobs index) in rice and barnyardgrass roots at different growth stages for OTUs constructed at 0.03 dissimilarity for bacterial and fungal sequences. Significant differences are indicated by different letters.

**Figure 2 plants-11-01592-f002:**
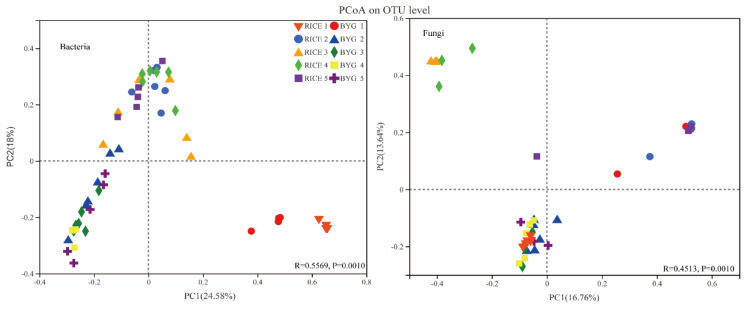
Principal coordinate analysis (PCoA) based on Bray–Curtis dissimilarity between the rice and barnyardgrass for endophytic bacteria and fungi in roots at different growth stages. PCoA of the unweighted UniFrac distance matrix representing differences in community structure of rice and barnyardgrass microcosms at five growth stages. Note: RICE1-5 and BYG1-5 represent the rice samples and barnyardgrass samples, collected from five different growth stages (BBCH 17, 24, 37, 45, and 57).

**Figure 3 plants-11-01592-f003:**
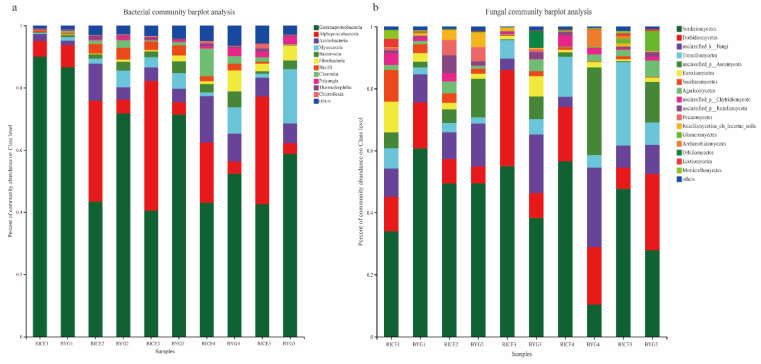
Endophytic bacterial (**a**) and fungal (**b**) community structure at the class level in the rice and barnyardgrass roots among five different growth stages.

**Figure 4 plants-11-01592-f004:**
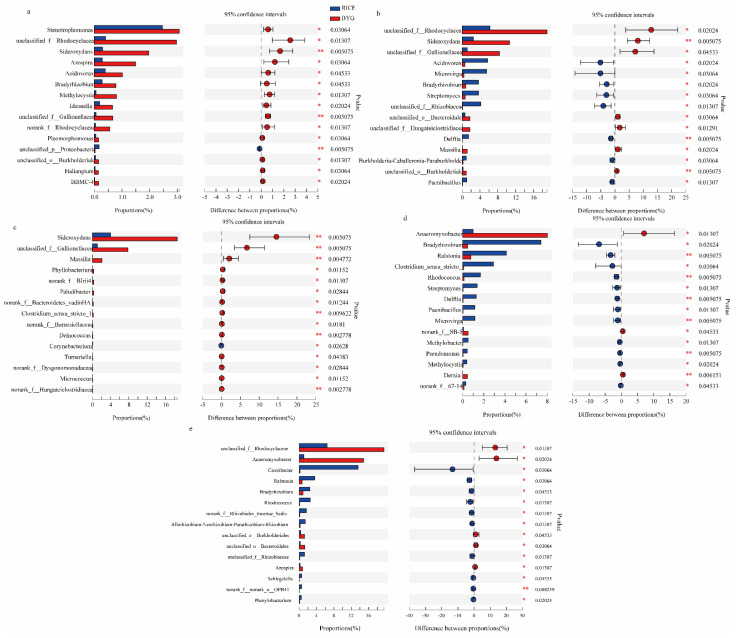
Endophytic bacteria were significantly different between rice and barnyardgrass groups at the genus level at BBCH 17 (**a**), 24 (**b**), 37 (**c**), 45 (**d**), and 57 ©.

**Figure 5 plants-11-01592-f005:**
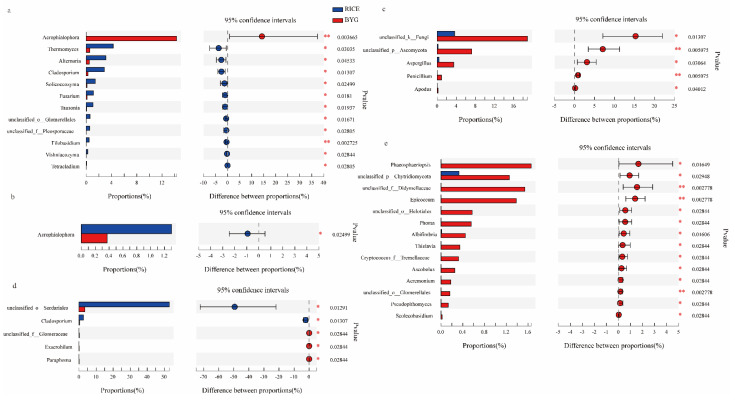
Endophytic fungi were significantly different between rice and barnyardgrass groups at the genus level at BBCH 17 (**a**), 24 (**b**), 37 (**c**), 45 (**d**), and 57 €.

**Figure 6 plants-11-01592-f006:**
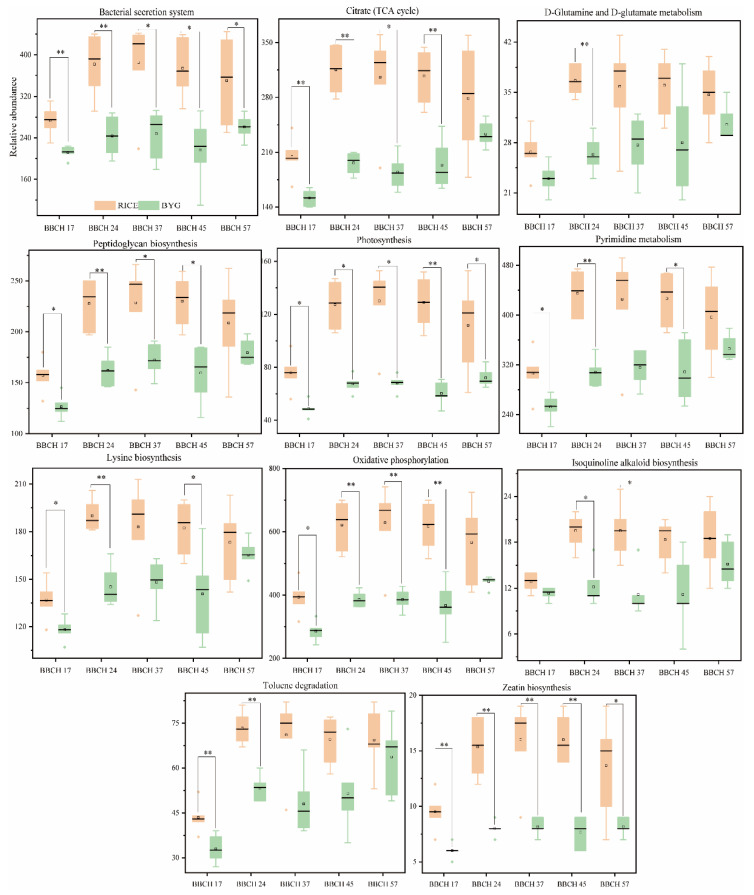
The significant microbial functional features in rice. Significant differences are indicated by asterisks. * *p* < 0.05, ** *p* < 0.01.

**Figure 7 plants-11-01592-f007:**
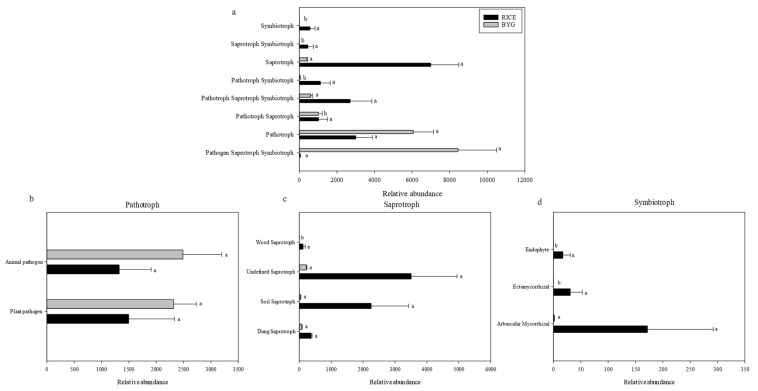
The relative abundance of trophic modes (**a**) and Guilds (**b**–**d**) assigned by FUNGuild for fungal communities. Different letters indicate significant difference between treatments detected by Tukey’s multiple test (*p* < 0.05).

**Figure 8 plants-11-01592-f008:**
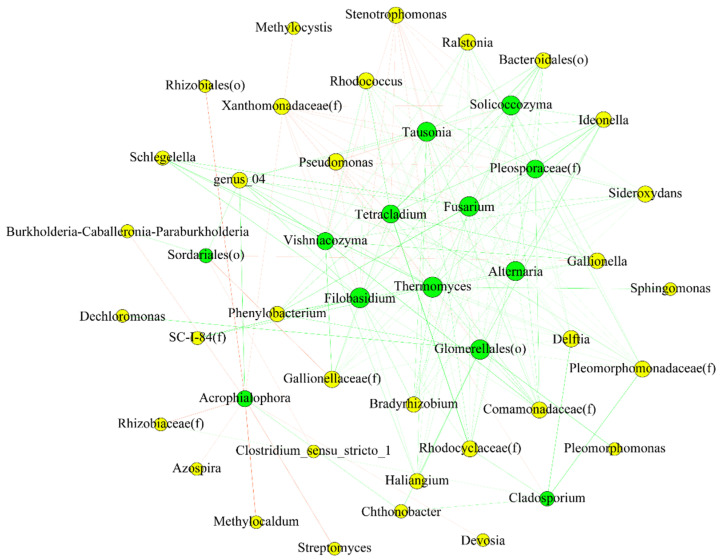
Correlation network diagram of bacteria and fungi at the genus level in rice.

**Figure 9 plants-11-01592-f009:**
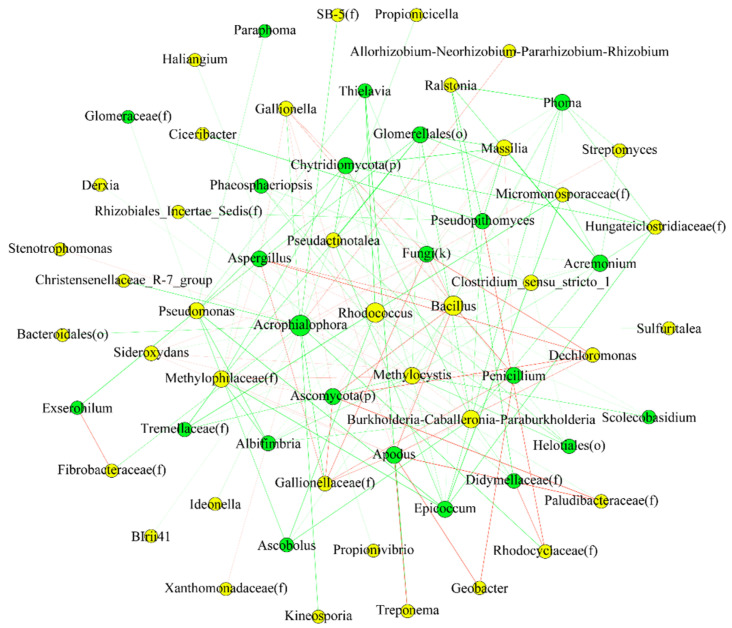
Correlation network diagram of bacteria and fungi at the genus level in barnyardgrass.

**Table 1 plants-11-01592-t001:** Species diversity (Shannon index), richness (Chao index), and coverage (Coverage index) soil samples for OTUs constructed at 0.03 dissimilarity for bacterial and fungal sequences.

	Sample	Shannon Index	Chao Index	Coverage Index
Bacteria	soil 1	2.97 ± 0.07 a	144.60 ± 62.58 a	0.18 ± 0.10 a
	soil 2	2.95 ± 0.01 a	131.00 ± 34.64 a	0.16 ± 0.01 a
	soil 3	2.92 ± 0.01 a	120.00 ± 40.99 a	0.20 ± 0.04 a
Fungi	soil 1	3.13 ± 0.22 a	515.43 ± 44.59 a	0.99 ± 0.00024 a
	soil 2	3.37 ± 0.04 a	542.99 ± 6.77 a	1.00 ± 0.00004 a
	soil 3	3.36 ± 0.02 a	528.37 ± 7.63 a	1.00 ± 0.00013 a

Note: data are represented as mean ± standard error (S.E.). Statistical analyses were performed with the Mann–Whitney U test between the two groups. The richness estimator (Chao), diversity estimator (Shannon), and coverage estimator (Coverage) were calculated at 3% distance. *p* < 0.05. Variants labeled with the same letters are not significantly different. Soils 1–3 stand for soil collected at three different spots from the same farm field.

## Data Availability

Not applicable.

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
