# Peer review of "Endophytic Fungal and Bacterial Microbiota Shift in Rice and Barnyardgrass Grown under Co-Culture Condition"

_plants, 2022, doi:10.3390/plants11121592_

Round 1
Reviewer 1 Report
The manuscript plants-1750207 evaluated the shift in the endophytic fungal and bacterial communities of Oryza sativa in the presence of Echinochloa crus-galli co-culture. Investigations were performed by16S rRNA and ITS metabarcoding and bioinformatic approaches. The manuscript is interesting and could add knowledge to the field. The authors carried out the experiments with a valid experimental design and used suitable sample processing and data handling techniques. The major limitation of the manuscript is the English language and figures quality which needs substantial improvements.
Broad comments
Introduction: The introduction places the study in a broad context and highlights why the study was needed. The working hypotheses, the aim(s), and the methodologies of the study are presented.
Materials and Methods: The description of the methods lacks some details.
· L72 – Provide more information on seed used and where they were retrieved.
· L81 – Was the soil subjected to treatments before being used in the trial? If yes state when and how the soil sample was sampled for analysis.
· L119 – Add primers used for ITS analysis.
· L150-152 – Add more details for these bioinformatic approaches (e.g., data filtering, access date websites, matrices transformation to get nodes and edges network analysis and software/packages employed).
Results: The results section is almost clear. The figures are not appropriate, their quality must be enhanced. Authors could also present the results of figures 1 and 2 as tables.
Discussion: The discussion is well structured.
Conclusions: Future perspectives in the conclusion section must be enhanced.
Other comments
I would change “genuses” term with “genera” which is more used and accepted.
In supplementary material define others (unknown and uncultured or other taxa excluded by filtering cut off?).
Reviewer 2 Report
The authors are advised to improve the manuscript in terms of adequate language levels as well as research paper structure.
I recommend expanding: The introduction section with more updated literature
Many sentences do not have the correct punctuation, and it is difficult to read the text.
English should be improved; grammar needs enhancement in many sentences and paragraphs.
All figures need for resolution enhancement.
Figures is not in printable quality. Also, some portions of the texts are losing their readability while sizing the image as per the text area. Kindly provide a better quality figure and increase the size of the figures.
Please check the References in-text and end-list for uniformity in style.
The conclusion you have provided is quite brief and provides sufficient feedback on the main objectives of your study.
Round 2
Reviewer 1 Report
Thank you for the manuscript revision and responses to my previous comments. I have no further suggestions.